# Effect of Amyloid-β Monomers on Lipid Membrane Mechanical Parameters–Potential Implications for Mechanically Driven Neurodegeneration in Alzheimer’s Disease

**DOI:** 10.3390/ijms22010018

**Published:** 2020-12-22

**Authors:** Dominik Drabik, Grzegorz Chodaczek, Sebastian Kraszewski

**Affiliations:** 1Laboratory of Cytobiochemistry, Faculty of Biotechnology, University of Wrocław, F. Joliot-Curie 14a, 50-383 Wrocław, Poland; 2Łukasiewicz Research Network-PORT Polish Center for Technology Development, Stabłowicka 147, 54-066 Wrocław, Poland; Grzegorz.Chodaczek@port.lukasiewicz.gov.pl; 3Department of Biomedical Engineering, Faculty of Fundamental Problems of Technology, Wrocław University of Science and Technology, Pl. Grunwaldzki 13, 50-377 Wrocław, Poland; Sebastian.Kraszewski@PWr.edu.pl

**Keywords:** membrane mechanics, molecular dynamics, flicker-noise spectroscopy, neurodegeneration, amyloid-beta peptides, pressure wave, giant unilamellar vesicles

## Abstract

Alzheimer’s disease (AD) is a neurodegenerative disease that results in memory loss and the impairment of cognitive skills. Several mechanisms of AD’s pathogenesis were proposed, such as the progressive accumulation of amyloid-β (Aβ) and τ pathology. Nevertheless, the exact neurodegenerative mechanism of the Aβ remains complex and not fully understood. This paper proposes an alternative hypothesis of the mechanism based on maintaining the neuron membrane’s mechanical balance. The incorporation of Aβ decreases the lipid membrane’s elastic properties, which eventually leads to the impairment of membrane clustering, disruption of mechanical wave propagation, and change in gamma oscillations. The first two disrupt the neuron’s ability to function correctly while the last one decreases sensory encoding and perception enabling. To begin discussing this mechanical-balance hypothesis, we measured the effect of two selected peptides, Aβ-40 and Aβ-42, as well as their fluorescently labeled modification, on membrane mechanical properties. The decrease of bending rigidity, consistent for all investigated peptides, was observed using molecular dynamic studies and experimental flicker-noise techniques. Additionally, wave propagation was investigated with molecular dynamic studies in membranes with and without incorporated neurodegenerative peptides. A change in membrane behavior was observed in the membrane system with incorporated Aβ.

## 1. Introduction

Alzheimer’s disease (AD) is an irreversible neuropathological disease with a progressive loss of the structure and function of neurons that slowly leads to memory and cognitive skills impairment [1]. AD is considered the most common cause of dementia, with increasing age being the most significant factor for AD occurrence. Many aspects of AD’s pathophysiology have been investigated and understood. However, several knowledge gaps still exist [2]. At the microscale, the AD’s brain is characterized by the presence of both amyloid plaques and neurofibrillary tangles [3]. Amyloid plaques display a broad range of morphological and biochemical characteristics and contain numerous proteins, mostly amyloid-β (Aβ). It is generally believed that AD’s pathogenesis is related to alternations in APP (Aβ precursor protein) processing, which results in the progressive accumulation of Aβ protein [4]. The most common form of this protein is 40 amino acids long and is called Aβ-40. Less common, yet believed to be associated with the disease, is the most hydrophobic and toxic peptide isoform, Aβ-42 [2]. Due to its physical characteristics, Aβ often acquires the configuration of β-pleated sheets and shows a greater tendency to aggregate, forming the core of the amyloid plaque composed of Aβ oligomers and fibrils. It is known that clearance abnormality leads to the accumulation of Aβ in the brain and central nervous system. Despite several studies showing the neurotoxicity of various forms of Aβ, the mechanism through which Aβ monomers, oligomers and other APP metabolites might lead to synaptic damage and neurodegeneration is not completely clear [5]. Several possibilities are under investigation, including alternation in signaling pathways related to synaptic plasticity, neuronal cell death, neurogenesis, and ion homeostasis disruption [6,7]. On the other hand, recent clinical trials showed that the elimination of Aβ does not affect the progression of AD [8]. To this end, the focus has shifted to the tau (τ) protein as a secondary pathogenic event that causes neurodegeneration [9]. While this is important for developing therapeutic strategies, it does not change the fact that Aβ proteins are mainly responsible for the further development of AD.

As stated previously, plaques’ major components are the Aβ peptides derived from the APP proteolytic processing at lipid membrane domains [10]. The membrane’s ability to dynamically cluster its components regulates the spatial and temporal assembly of signaling and trafficking molecules. The formation of short‑lived signaling platforms can be vital in this case [11,12]. These platforms can be classical sphingolipid-cholesterol ordered domains, ceramide‑rich platforms, or other areas with different biophysical properties [13]. For instance, gangliosides’ presence was reported to increase the incorporation of Aβ-42 [14]. This is especially important for cell signaling, axon sorting and guidance, neural development, and synaptic plasticity [15]. Furthermore, microdomains in neurons are required to maintain dendritic spines and healthy synapses making them essential for neural communication, memory, and learning [16]. A membrane, to preserve the ability to form such microdomains (signaling platforms) spontaneously, would require a certain mechanical balance. The disruption of such a delicate balance could, in this case, lead to the slow and progressive loss of membrane functions [17]. Interestingly, the composition of detergent-resistant membranes purified from AD brains is abnormal–they are more ordered and more viscous [18]. It was also shown that neurons in AD have a significantly different membrane composition, including a lower level of sphingomyelin and a higher level of ceramides [19,20], which are known to significantly alter mechanical properties. This implies a potential disturbance of mechanical balance. Furthermore, it was recently suggested that the Hodgkin Huxley model of nerve propagation, based on local ion current flow, does not fully explain how membrane potential cause the opening and closing of the ionic channels. Based on the propagation of pressure waves and membrane mechanical properties, a complement model hypothesis was proposed [21,22].

In this paper, we decided to investigate whether Aβ monomers’ accumulation in the membrane disrupts the membrane’s mechanical balance. We focused on measuring the mechanical changes of POPC membranes as they can be used as a starting point for discussion or more complex approaches with neuron-mimicking membranes. We investigated the effect of Aβ monomers on mechanical properties, mainly focusing on the bending rigidity coefficient. These parameters were measured using flicker-noise spectroscopy, which links spontaneous bilayer fluctuations with its mechanical properties–and Molecular Dynamics simulations of whole lipid vesicles. Furthermore, we have simulated the pressure wave propagation on membranes with and without Aβ peptide to investigate the latter’s effect on membrane behavior. Even if the proposed study is not thoroughgoing research, we aimed to draw attention to another vital aspect, namely mechanically-driven molecular phenomena during neurodegeneration in AD.

## 2. Results & Discussion

### 2.1. Effect of Aβ Peptides on Structural Parameters

Table 1 presents the calculated basic structural properties of a membrane with incorporated different Aβ monomers. Snapshots of simulated systems are presented in Figure 1. Both membrane thickness and area per lipid (APL) of vesicles with incorporated peptides differed with statistical significance from other populations. Specifically, one-way ANOVA reported the difference between the means of membrane thickness and APL as statistically significant. The following post-hoc Tukey test reported the difference to be significant between all investigated populations. It also should be noted that the difference of the POPC population’s parameters from other Aβ populations’ parameters is especially conspicuous. The membrane thickness was higher in vesicles with Aβ, suggesting that their presence contributed to the elevation of either whole lipid molecules or just phosphorus atoms in the bilayer due to bilayer remodeling. While APL was lower when compared to the base system, this result is not surprising as additional particles in the bilayer contributed to the more tightly packed conformation. These results somewhat contradict the literature, as it was suggested that binding of Aβ peptide to the membrane might result in both compression of the bilayer (higher APL) and making it thinner (lower membrane thickness) [7]. On the other hand, it was reported that both membrane properties (i.e., lipid bilayer thickness) regulate the generation and surface-induced aggregation of Aβ peptides and the incorporation of Aβ peptides induces membrane remodeling, such as elevation of lipids in the vicinity of the peptides [23]. Figure 2 shows the bilayer profiles of systems with different Aβ peptides. Interestingly, the peptides are localized in the whole interphase region. This positioning in the lipid bilayer was in agreement with the literature, as Aβ peptides in monomeric form have been reported to localize in the intermediate region between the lipid head group and acyl chains. Only when the dimer was formed did it slowly submerge deeper into the bilayer, eventually reaching the transmembrane state [24]. All of the investigated vesicles sustained their quasi-spherical geometry.

### 2.2. Effect of Aβ Peptides on Bending Rigidity Coefficient

The bending rigidity coefficient was determined using both computational and experimental techniques. These results are presented in Table 1. Power spectra of investigated MD, which were used for bending rigidity determination, are presented in Figure 3. The analysis of these systems showed that the bending rigidity of vesicles with Aβ peptides was almost twice as low as in the reference POPC system. However, there was no difference in mechanical properties between the systems with incorporated peptides. These results are in agreement with our experimental results. When measured using flicker­noise spectroscopy, the bending rigidity differed with statistical significance between the populations using both tests: ANOVA and Kruskal–Wallis. The post-hoc Tukey test showed that the difference only occurs between the reference POPC vesicle and the systems with Aβ peptides. There was no statistically significant difference between systems with Aβ monomers. Results were similar for both statistical (SA) and average-based (AVB) approaches. More details are presented in Appendix A of Appendix A. This decrease of membrane bending rigidity due to the presence of Aβ peptides is in agreement with the literature [26]. This effect was reported to be more robust in the case of oligomers and fibrils, but was still present in the case of the peptides. Aβ-42 was also reported to decrease Young’s modulus in neural cells [27]. Furthermore, it was reported that Aβ oligomers are inducing neural elasticity changes [28]. These results and references are consistent with our hypothesis stating that Aβ peptides do influence the membrane’s mechanical properties. Moreover, it was recently shown that remodeling of the membranes occurs after the incorporation of Aβ peptides [23]. However, this phenomenon was only observed in bilayers with low or medium bending rigidity (such as POPC) and not in membranes with higher bending rigidity (DLPC-1,2-dilauroyl-sn-glycero-3-phosphocholine). Combining the above with our results suggests that incorporating Aβ, which decreases the bending rigidity, could lead to further progression of membrane modeling and local disruption of membrane topology. This could result in the disruption of mechanical balance that could halve the ability to form spontaneous signaling platforms. While it was argued that Aβ incorporation is mostly electrostatically driven [7,29], it should also be noted that remodeling of membrane occurred in membranes with lower bending rigidity. It could somewhat explain why the likelihood of neurodegenerative disease occurrence increases with age. Aging of neural cells–aging of cells in general–does not tend to change their electrochemical potential but is known to change their mechanical properties [30]. Additionally, the progressing decrease of the bending rigidity (caused by aging) might influence the curvature of the lipid bilayers and/or induce packing defects, resulting in the exposure of hydrophobic clefts in the vesicle surface. Both factors were reported to promote the interaction of Aβ with the bilayer and its aggregation properties as well [31,32]. It should be noted that such a change in lipid membrane mechanical properties is especially relevant in neurons. It can modify both elasticity and viscosity, which influences signal transduction, leading to the loss of synaptic plasticity and impairment of neuronal signal propagation [33]. They can directly affect transmembrane proteins’ functioning, sever metabolic pathways, and disrupt transport between the membrane itself.

### 2.3. Effect of Aβ Peptides Pressure Wave Propagation

Finally, we investigated the effect of Aβ monomers’ presence in the membrane on propagation of the mechanical wave. According to Barz et al. [21] a mechanical (pressure) wave is necessary for the neuronal membrane to trigger ion pumps. To this end, two systems were subjected to the effect of pressure wave induced by water slab velocity change. One of the systems had Aβ monomers incorporated. The second did not. A snapshot and evolution in time of the system with incorporated peptides is presented in Figure 4. The evolution of amplitude of the wave observed on the membrane is presented in Figure 5B. It can be clearly seen that the amplitude of the wave is oscillating, reaching local maximum and minimum values alternately. However, a significant difference between the system with incorporated peptides and the control system can be observed. The maximum values of amplitude occur significantly later in the control system. Furthermore, in the system with Aβ peptides, additional increases in the amplitude were observed that were not seen in the control system (see Figure 5C). This strongly suggests that Aβ peptides’ presence influences the wave propagation and the membrane’s response to pressure waves. If a sufficient number of peptides were incorporated, this could differ the membrane’s response to disrupt nerve impulse propagation by inducing pump response too fast. Additionally, it could alter the membrane fluctuations to a point where a shift in membrane resonance frequency would occur. It was recently hypothesized that gamma neural oscillations (30–55 Hz) are responsible for sensory encoding and perception enabling [34]. Changes in gamma oscillations can be observed in several neurodegenerative diseases [35,36]. Interestingly, membrane properties strongly determine the characteristics of emergent gamma oscillations [37] and brain stimulation with gamma oscillations was reported to improve spatial and recognition memory during AD [38]. Our results suggest a very strict dependency between the lipid membrane’s mechanical properties and the resonance frequency and behavior of the membrane. Moreover, the results strongly support the alternative hypothesis of nerve propagation based on membrane mechanical properties and pressure wave propagation.

## 3. Materials and Methods

### 3.1. Materials

Lipid POPC (1-palmitoyl-2-oleoyl-glycero-3-phosphocholine) was purchased from Avanti Polar Lipids (Alabaster, AL, USA). Fluorescent probe Atto488-DOPE was purchased from Atto-Tech (Siegen, Germany). Peptides β-Amyloid 1–40 and β-Amyloid 1–42 (Human) were purchased from Sigma-Aldrich (Poznań, Poland). Labelled peptide β-Amyloid 1-40-TAMRA was purchased from Eurogentec (Seraing, Belgium). Detergent DOTM (Decyl-B-D-1-Thiomaltopyranoside) was purchased from Sigma Aldrich. Bio-Beads SM-2 were purchased from Bio-Rad (Warszawa, Poland). PBS tablets were purchased from VWR (Gdańsk, Poland).

### 3.2. Preparation of Giant Unilamellar Vesicles (GUVs)

A modified electroformation method was used to enable the incorporation of Aβ peptides [39]. Briefly, 20 μL of chosen lipid and detergent mixture in chloroform were deposited in small quantities (as 2 μL droplets) onto platinum electrodes. The concentration of lipid POPC was 1 mg/mL, while the concentration of DOTM detergent was calculated so that its final concentration in the electroformation chamber was equal to 75 μM. Two electrodes were set parallel to one another at a distance of 5 mm. The electrodes were kept for 1 h under reduced pressure to remove traces of organic solvents. Next, the electrodes were immersed in 400 mM sucrose solution. This was followed by applying AC voltage to electrodes with 1 Hz frequency and 1 V amplitude. The voltage was increased by 1 V every hour up to 4 V [40]. After the electroformation, chambers were left for 1 h without an electrical field applied to allow the descent of vesicles from electrodes. This was followed by buffer exchange and peptide incorporation. Drops of GUVs solution (50 μL) were transferred to 100 μL of 1.5 × PBS solution with 75 μM DOTM. The peptides were dissolved in 1% NH_4_OH solution and sonicated for 30 s. This was followed by the addition of 5× PBS buffer to obtain 1× PBS. The peptides were added to GUVs/PBS solution in such quantity to obtain 10 m% with respect to lipids and incubated for 12 h at room temperature. After that, DOTM removal was carried out using BioBeads in two batches. More detailed verification of Aβ monomers incorporation in GUVs can be found in Appendix A.

### 3.3. Confocal Microscopy Imaging and Acquisition

A Cell Observer SD spinning disk confocal microscope (Zeiss, Jena, Germany) equipped with a Plan-Apochromat 100×/1.46 oil immersion objective (Zeiss) was used for vesicle recording. 512 × 512 pixels images were recorded with an EMCCD camera (Rolera, QImaging, British Columbia, Canada) using 2 × 2 binning with 0.133 μm pixel size with a video integration time of 30 ms. At least 5000 images were recorded for each vesicle. Samples were illuminated with 488 nm laser and emitted light passed through 527/54 filter. All samples were measured at 23 ± 1 °C. All measurements have been performed in a dedicated PTFE observation chamber with 300 μm height to reduce the effect of uncontrolled vesicle movements. The value of the depth of focus was equal to 0.85 μm. To enhance the quality of analysis the radius of a vesicle was calculated for each image. In case the change of radius was considered as an outliner, the image in the series was discarded from further analysis. It occurred due to misdetection caused by noise or other reasons described in previous work [41].

### 3.4. Flicker-Noise Spectroscopy Analysis

The flicker-noise spectroscopy technique is based on the analysis of a vesicle shape fluctuations over time. It is used to determine the bending rigidity coefficient from those fluctuations using the Helfrich’s theory. Measurements were performed following our established protocol [41]. Briefly, the membrane fluctuation spectrum was extracted from every single recorded image of the same lipid vesicle using custom software. To calculate the bending rigidity coefficient from a set of time-lapsed two-dimensional images a correlation with three-dimensional membrane elasticity model was established. This was achieved by means of the angular autocorrelation function. The bending rigidity coefficient κ and membrane tension σ can be determined using two approaches, namely the statistical [41,42] and the average-based approach [41,43]. While the average-based approach might seem more straightforward, the main advantage of the statistical one is the histograms that show the characteristics of vesicle fluctuations and are in agreement with the model.

### 3.5. Molecular Dynamics Simulations

Molecular dynamics (MD) has been effectively used as a tool for studies of Aβ structures [44]. However, due to a high computational cost, many of these studies are performed on shorter Aβ segments in order to decrease the system size. Nevertheless, the MD simulations are used to study Aβ structures such as peptides [45], dimers [46] and oligomers [47]. In our work, we adopted a structure of Aβ peptides presented by Crescenzi et al. [48]. The full-atomistic MD simulation was performed using NAMD 2.9 [49] software with CHARMM36 united-atom force field [50] under NPT conditions (constant: Number of particles, Pressure and Temperature). Two different types of simulation were carried out, namely peptide incorporation on planar bilayer simulations and lipid vesicle simulation.

For peptides on planar bilayer simulations: a planar POPC membrane system consisting of 200 lipid molecules (100 on each of leaflets) was used. To determine peptide docking in the bilayer, several simulations were performed with various placement of the investigated peptide on the bilayer. The simulation was run till the peptide was incorporated into the bilayer and remained incorporated for at least 30 ns or it was not incorporated into the bilayer. Successful simulations for Aβ-40, Aβ-42, and Aβ-40 TAMRA were carried out for 62, 46, and 51 ns, respectively. The obtained stable position of the investigated peptide in the bilayer was later used for its manual incorporation in the vesicle system. More detailed information about systems setup and properties is presented in Appendix A.

For lipid vesicle simulations: a POPC vesicle was modelled as a liposome of 20 nm radius both sides hydrated with TIP3P water molecules, giving a final simulation box of 30 nm^3^. The vesicle system was selected over the planar system, as it differs in curvature, which can significantly change attached peptide’s behavior [51]. Furthermore, we showed that, in some cases, results from vesicles systems agreed with the experimental data, while for the planar system results were completely different [25]. Three-dimensional periodic boundary conditions were applied to deal with potential energy disruption due to the origin cell discontinuity. The vesicle system was created using a custom script in Matlab. Starting APL was set as 68.1 on average [52], but was corrected accounting for the effect of vesicle’s curvature. The APL value was multiplied by 0.95 for the inner and by 1.05 for the outer leaflets, respectively. The vesicle system was equilibrated prior to the addition of Aβ peptides for 100 ns. This was followed by peptide incorporation, which was done using a custom script. Peptides were equally distributed on the vesicle, merged with the liposome system and solvated. After peptide incorporation vesicles were equilibrated for additional 20 ns, followed by running for at least 10 ns and then analyzed. To determine the stable equilibration time-point, six selected parameters (vesicle radius, the thickness of lipid bilayer, mean values and standard deviations of both inner and outer leaflets) were continuously monitored. More detailed information about systems setup and properties is presented in Appendix A.

For pressure wave propagation simulations: simulations were performed according to the established procedure [53,54]. Specifically, the planar system with incorporated β peptides was multiplied 4 times with an additional water slab in Z-axis. After equilibration of the membrane system was switched from NPT to NVE (constant: Number of particles, Volume and Energy) conditions. The pressure wave was modelled as the momentum change of water particles in Z-axis by averaged velocity Δvz defined by Equation (1), where I is pressure impulse, A denotes the area of changed water particles, m the mass of water, and N the number of changed water particles.
(1)$$\Delta v_{z}=\frac{I\cdot A}{m\cdot N}$$

The simulated pressure wave was equal to 10 μN/m^2^·s (1 mPa·s). An evolution of bilayer bending and return to equilibrium was investigated. The position of phosphorus atoms was used to bin membrane position in the OX plane. The obtained bending characteristic of membrane was fitted with the sine function to parametrize the system’s behavior and evolution in time.

### 3.6. Determination of Bending Rigidity Coefficient in MD

To determine the bending rigidity of model lipid vesicles, we adopted an algorithm originally developed by Braun & Sachs [25,55]. It has an advantage over other approaches [56,57,58] as it determines mechanical properties based on fluctuations of the bilayer within the vesicle, which can be different than for planar lipids [25]. In short, each lipid is described by a vector spreading from the head (phosphorus atom) up to tail position (midpoint of both 16th carbon atoms in each of tails). This is followed by the discrete surface representation θ, φ using a grid. For each time-point, the surface of fluctuations is established by detecting of fitted sphere’s origin point, converting bilayer fluctuations into spherical coordinates and subtracting the radius value. Finally, the average of both inner and outer leaflets fluctuations is calculated. This is followed by spectral harmonics analysis (SPHA) for calculated fluctuations. Eventually, the Helfrich’s approach is used by establishing spherical harmonic coefficients $a_{lm}$. The obtained $a_{lm}$ undulation power spectrum can be interpreted according to the Helfrich continuum model for undulations on a sphere with vanishing spontaneous curvature.

### 3.7. Determination of Basic Structural Parameters

Additionally, basic structural parameters were determined from performed MD simulations to establish the effect of Aβ peptides incorporation. These include membrane thickness, area per lipid and vesicle density profiles. For each frame position a sphere fit to phosphorus atoms in inner leaflet, in outer leaflet and to both was done in order to obtain radius for inner leaflet, for outer leaflet and for whole vesicle, respectively. Membrane thickness is calculated as a difference between the radius of outer and inner layers. The area per lipid for the whole vesicle was calculated according to Equation (2) using Braun and Sachs approach [55].
(2)$$ALP_{vesicle}=\frac{4\pi r_{vesicle}^{2}}{\frac{1}{2}\left(n_{L,inner}+n_{L,outer}\right)}$$

To determine density vesicle profiles, three crucial zones of each vesicle area were selected: Head-groups, Carbonyl-Glycerol, and Acyl-Chain, respectively. The distance from radius was calculated for each particle and then histogrammed. Obtained results were followed by the normal distribution fit.

### 3.8. Statistics

To test for the significant difference between the parameters, one-way ANOVA with the significance level set at 0.05 was used for presenting the normal distribution of data. The Tukey test was used as a post hoc test. The Kruskal–Wallis test with the significance level set at 0.05 was additionally used for data with the risk of not the following a normal distribution. All statistical analysis were performed using the OriginPro 2015 (OriginLabs) software. Averaged values are presented with standard deviation values.

## 4. Conclusions

In this paper, we investigated the effect of Aβ monomers on the mechanical properties of the POPC vesicle. Additionally, we have investigated the change in basic membrane parameters, such as membrane thickness and APL. We showed that membrane thickness increases and APL decreases after the incorporation of Aβ. We postulate that this results from the bilayer’s mechanical remodeling after Aβ monomer incorporation reported previously in the literature. We observed a decrease in the bending rigidity coefficient after incorporating Aβ in both MD simulation and the flicker-noise experiment. Since it was reported in the literature that Aβ-induced membrane remodeling is more likely when bending rigidity is smaller, we believe that the progressing decrease of bending rigidity might be considered a driving factor of neurodegeneration progression. We managed to prove that incorporating Aβ peptides influences the mechanical properties of lipid membranes, suggesting that membrane’s mechanical properties may play a more important role in neurodegenerative disorders. Finally, we have investigated the difference in wave propagation on membranes with and without incorporated Aβ peptides. We showed that the presence of the peptides did change the behavior of the system. This could lead to an impediment of nerve impulse propagation and/or shift in membrane resonance frequency and, as a result, disrupt gamma oscillations. In summary, this paper aims to draw attention to an important hypothesis that mechanically driven molecular phenomena that originate from the membrane could be a critical factor in the pathogenesis of AD. We believe that we presented satisfactory results to support this claim. We acknowledge that this study was limited to the effect on POPC lipid membrane only. Further studies should be focused on more biologically relevant membranes, specifically on the disruption of their ability to cluster dynamically. The effect of mechanical changes on lipid metabolism/signaling should also be addressed in future studies.

## Figures and Tables

**Figure 1 ijms-22-00018-f001:**
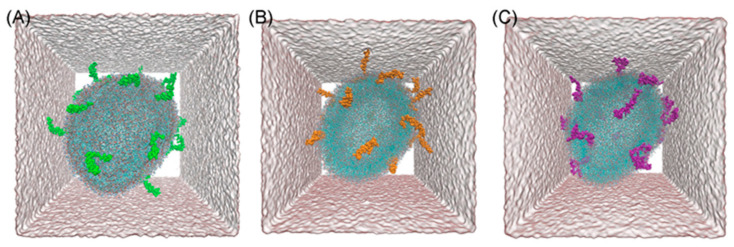
Snapshots of POPC vesicle systems with incorporated 10 m% Aβ monomers: (**A**) Aβ-40, (**B**) Aβ-42 and (**C**) Aβ-40-TAMRA.

**Figure 2 ijms-22-00018-f002:**
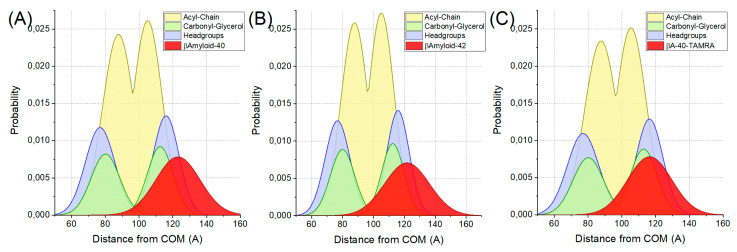
Probabilities of specified membrane regions for POPC vesicles with (**A**) Aβ-40, (**B**) Aβ-42 and (**C**) Aβ-40-TAMRA peptides incorporated in function of distance from center of mass. Specified regions are headgroups (blue), carbonyl-glycerol (green), acyl-chain (yellow) and peptides (red).

**Figure 3 ijms-22-00018-f003:**
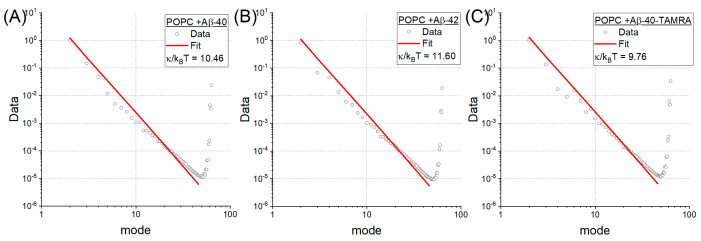
Power spectra along with model fits of investigated Molecular Dynamics systems with (**A**) Aβ-40, (**B**) Aβ-42 and (**C**) Aβ-40-TAMRA monomers, which were used for κ (bending rigidity) determination.

**Figure 4 ijms-22-00018-f004:**
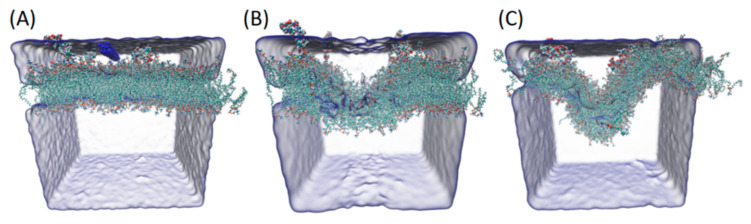
Snapshots of POPC system with incorporated Aβ peptides throughout pressure wave propagation. In panel (**A**) water particles with increased velocity are marked blue. In panel (**B**) a snapshot of system after 13 ps is presented. In panel (**C**) a final snapshot of simulation after 3 ns is presented.

**Figure 5 ijms-22-00018-f005:**
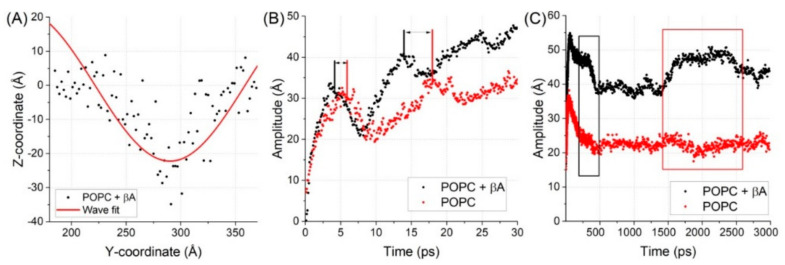
In panel (**A**) the fitting of sin function to histogrammed phosphorus atoms’ positions to determine the characteristics of the wave. In panel (**B**) and (**C**) evolution of the amplitude is presented in shorter and longer times of the simulations, respectively.

**Table 1 ijms-22-00018-t001:** Summary of Calculated Parameters from a Molecular Dynamics (MD) Study and Flicker Noise Spectroscopy Measurements for POPC membrane with incorporated Aβ (amyloid-β) monomers.

System Description	Membrane Thickness [nm]	APL of Vesicle [Å^2^]	κ (MD) [J]	κ (Flicker, SA) [J]	κ (Flicker, AVB) [J]
POPC [25]	$\left(3.43\pm 0.01\right)$	$\left(61.4\pm 0.1\right)$	$\left(7.3\right)\times 10^{-20}$	$\left(11.7\pm 6.9\right)\times 10^{-20}$	$\left(10.5\pm 2.2\right)\times 10^{-20}$
POPC with 10 m% Aβ-40	$\left(3.63\pm 0.01\right)$	$\left(56.4\pm 0.1\right)$	$\left(4.3\right)\times 10^{-20}$	$\left(3.7\pm 1.4\right)\times 10^{-20}$	$\left(3.6\pm 2.0\right)\times 10^{-20}$
POPC with 10 m% Aβ-42	$\left(3.65\pm 0.01\right)$	$\left(56.3\pm 0.1\right)$	$\left(4.8\right)\times 10^{-20}$	$\left(3.1\pm 1.9\right)\times 10^{-20}$	$\left(3.3\pm 2.1\right)\times 10^{-20}$
POPC with 10 m% Aβ-40-TAMRA	$\left(3.66\pm 0.01\right)$	$\left(56.0\pm 0.1\right)$	$\left(4.0\right)\times 10^{-20}$	$\left(3.4\pm 1.9\right)\times 10^{-20}$	$\left(3.1\pm 1.3\right)\times 10^{-20}$

APL, area per lipid; κ, bending rigidity coefficient; MD, molecular dynamics; AVB, average-based approach; SA, statistical approach.

## Data Availability

Most of the data is available in the manuscript supplementary information. The simulation data presented in this study are available on request from the corresponding author. This data are not publicly available due to GBs files sizes.

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
