# Peer review of "Effect of Amyloid-β Monomers on Lipid Membrane Mechanical Parameters–Potential Implications for Mechanically Driven Neurodegeneration in Alzheimer’s Disease"

_ijms, 2020, doi:10.3390/ijms22010018_

Round 1

Reviewer 1 Report

The original paper from Drabik et al., report a molecular dynamics study on the effect of Abeta species on membrane thickness and mechanics, and APL. They show that Abeta peptides increases membrane thickness, reduce bending rigidity, and wave propagation, possibly affecting signalling platforms and neuronal synaptic activity in AD.

The results are quite interesting and of value for the recent revision of the amyloid hypothesis.

Where the peptides used as monomer or oligomers? several studies already reported the effect of the oligomeric amyloid. This should be clearly described and discussed.

What about the effect on organelle membranes? and on lipids metabolism/turnover?

Introduction is definitively too long; some information can be moved to the discussion-conclusion sections.

It would be nice to acknowledge the original hypothesis formulated 15 years ago by Marchesi (Yale Univ.) on PNAS.

Author Response

The original paper from Drabik et al., report a molecular dynamics study on the effect of Abeta species on membrane thickness and mechanics, and APL. They show that Abeta peptides increases membrane thickness, reduce bending rigidity, and wave propagation, possibly affecting signalling platforms and neuronal synaptic activity in AD. The results are quite interesting and of value for the recent revision of the amyloid hypothesis.

Point 1: Where the peptides used as monomer or oligomers? several studies already reported the effect of the oligomeric amyloid. This should be clearly described and discussed.

Response 1: We thank the Reviewer for this comment. In this work, we are investigating the effect of monomers. We have made several corrections in the MS to emphasize that monomers are investigated.

Point 2: What about the effect on organelle membranes? and on lipids metabolism/turnover?

Response 2: The Reviewer is raising a very interesting aspect that can lead to future studies of cell membrane's mechanical changes in neurodegenerative diseases. Unfortunately, this study was limited only to the effects of Abeta monomers on POPC membranes. We have added limitations and future studies in the conclusions section which highlights the possibility of further investigations of these two directions in the future (page 9, line 344).

Literature results suggest that there will be an effect of mechanical changes on lipid metabolism. This can be already, not directly, found in literature where effect of lipid membrane type of vesicle rupture caused by PLD is investigated (Park 2011 J Membrane Biol, Park 2012 J Membrane Biol or Park 2012, Colloids and Surfaces B: Biointerfaces).

Point 3: Introduction is definitively too long; some information can be moved to the discussion-conclusion sections.

Response 3: We have made an attempt to shorten the Introduction section and moved some parts to the Results and Discussion section. We hope that the article’s structure is better in the corrected form.

Point 4: It would be nice to acknowledge the original hypothesis formulated 15 years ago by Marchesi (Yale Univ.) on PNAS.

Response 4: We thank for this remark. We have added the citation (on page 2, line 72, citation 17) to the Introduction section, as reviewer requested.

Reviewer 2 Report

I have reviewed manuscript entitled ''Perhaps neurodegeneration in Alzheimer disease is mechanically driven?''

Authors have focused on the major hallmark of AD pathophysiology. 

Overall, the study was good; however, the manuscript deserves some revisions.

The title could be modified. The authors may consider titled as per their findings. 

Limitations and future study directions are missing. How could this study be translated into clinical research? 

Author Response

I have reviewed manuscript entitled ''Perhaps neurodegeneration in Alzheimer disease is mechanically driven?'' Authors have focused on the major hallmark of AD pathophysiology. Overall, the study was good; however, the manuscript deserves some revisions.

Point 1: The title could be modified. The authors may consider titled as per their findings.

Response 1: We thank the reviewer for this comment. We have changed the MS title to better reflect findings in the paper. The MS is now entitled “Effect of amyloid-β monomers on lipid membrane mechanical parameters – potential implications for mechanically driven neurodegeneration in Alzheimer’s disease”.

Point 2: Limitations and future study directions are missing. How could this study be translated into clinical research?

Response 2: We have added limitations and future studies in the conclusions section of the MS. It states as follows: " We acknowledge that this study was limited to effect on POPC lipid membrane only. Further studies should be focused on more biologically-relevant membranes, specifically on the disruption of their ability to cluster dynamically. The effect of mechanical changes on lipid metabolism/signalling should also be addressed in future studies." (page 9, line 344)

The translation of our results into clinical research is rather difficult at this stage. Studying the effect of membranes’ mechanical properties on organism functioning is still a relatively new approach. While we have a certain idea of how given particles influence the mechanical properties of the membrane (cholesterol being the prominent example), we are unaware of any clinical approaches that would focus on rejuvenating the cell membranes' mechanical properties. In this case, the issue becomes even more complicated, as a clinical trial showed that the elimination of Aβ did not affect AD’s progression (Holmes et al, 2008, Lancet). However, it is unclear whether Aβ peptides were eliminated from cell membranes. On the other hand, brain stimulation with gamma oscillations was reported to improve spatial and recognition memory during AD (Martorell et al, 2019, Cell). Since gamma oscillations are strongly dependent on the membrane’s mechanical state, the article’s hypothesis might provide a more general explanation of this effect. Additionally, AD brain membranes are significantly altered, which suggests that eliminating the culprit is not sufficient. An additional treatment of the membranes may be required to allow the brain’s rejuvenation and reverse the progress of neurodegenerative diseases. On a more general approach, if further studies are pursued, they could explain why hydrogel scaffoldings in the spinal cord or brain injuries - due to mechanical mismatch - prevent new neurons from functioning properly. However, this study was limited only to the effects of Aβ monomers on POPC membranes, and additional studies are still required to provide answers to the mentioned issues.